

# Analysing binding stoichiometries in NMR titration experiments using Monte Carlo simulation and resampling techniques

Conrad Hübler

Institut für Organische Chemie, Technische Universität Bergakademie Freiberg, Freiberg, Saxony, Germany

## ABSTRACT

The application of Monte Carlo simulation and resampling techniques to analyse possible binding stoichiometries in NMR titration experiments is presented. Four simulated NMR titration experiments having complex species with 1:1, 2:1 and 1:2 stoichiometries were each analysed using a 1:1, 2:1/1:1, 1:1/1:2 and a 2:1/1:1/1:2 model as implemented in *SupraFit*. Each best-fit model was inspected using Monte Carlo simulation (MC), Cross Validation (CV) and a new protocol termed Reduction Analysis (RA). The results of the statistical post-processes were used to calculate characteristic descriptors that are the base of the judgment for both, the models and individual stability constants. The results indicate promising approaches to correctly identify 1:1, 2:1/1:1 and 1:1/1:2 models, however with some limitations in case of the 2:1/1:1/1:2 model. All simulations and post-processing protocols were performed with the newly presented *SupraFit*.

## INTRODUCTION

Supramolecular chemistry, the chemistry dominated by noncovalent interactions, is a highly interdisciplinary field. While the first steps are connected to synthetic chemistry (*Lehn, 1988*; *Cram, 1988*; *Pedersen, 1988*; *Hoss & Vögtle, 1994*), physicochemical methods became important to characterise and understand supramolecular systems (*Schalley, 2012*). Nowadays, the role of supramolecular chemistry in biological processes, including pathogenic ones, has become more and more clear (*Mazik, 2009*; *Kubik, 2009*; *Mazik, 2012*).

The understanding of biological processes which are dominated by noncovalent interactions benefits from the analysis of smaller model systems, which can be comprehensively characterised with modern experimental (*Walker, Joshi & Davis, 2009*; *Mazik & Geffert, 2011*; *Francesconi et al., 2013*; *Rosien, Seichter & Mazik, 2013*; *Lippe & Mazik, 2013*; *Kaufmann et al., 2014*; *Lippe & Mazik, 2015*; *Traulsen et al., 2015*) and theoretical methods (*Wendler et al., 2010*; *Grimme et al., 2010*; *Caldeweyher et al., 2019*; *Hohenstein & Sherrill, 2012*; *Sure, Antony & Grimme, 2014*; *Antony, Sure & Grimme, 2015*;

Corresponding author
Conrad Hübler,
Conrad.Huebler@gmx.net

*Brauer et al., 2016*) as well as the combination of thereof (*Kaufmann et al., 2012*; *Lohse et al., 2015*). An elementary process is the complex formation between two components A and B (also termed host and guest). As a result of this complex formation, species of the various composition or stoichiometry like AB, $A_2B$, $AB_2$ or $A_aB_b$ in general may exist. These species are connected through individual stability constants which are usually of interest in the analysis of supramolecular systems. Assuming a system of A and B, where AB exist alongside $AB_2$, two individual stability constants $K_{11}$ and $K_{12}$ describe the ratio of the concentration of the various species:

$$K_{11} = \frac{[AB]}{[A][B]} \tag{1}$$

$$K_{12} = \frac{[AB_2]}{[AB][B]} \tag{2}$$

Common experimental techniques such as NMR titration or ITC experiments estimate the stability constants alongside other mandatory parameters (chemical shifts, heat of formation) based on the change of a signal that itself is a consequence of the change in the concentrations. The change in concentration is due to the variation of the initial concentration of the components A and B as set up during the experimental design. Typically, the concentration of one component (A) is kept fixed while the concentration of the other component (B) is varied resulting in a continuous change of the ratio of $[B]_0$ and $[A]_0$. As the underlying mass-balance equations to calculate the equilibrium concentrations are polynomial, the stability constants alongside with the other parameters are obtained after nonlinear curve fitting of the experimental data (titration curve) (*Thordarson, 2011*; *Thordarson, 2012*; *Hirose, 2006*). Given this fact, several difficulties arise: (a) The binding stoichiometry is the model input and not the output. Hence, the stoichiometry does not change during fit process, however parameters may become meaningless. (b) Nonlinear regression with multiple parameters can have many different solutions, of which some may have parameter with meaningless values after fitting. (c) In multiple linear regression, variable selection on the basis of statistical tests is well described in the literature, however in case of nonlinear regression a general or unified approach is to the best of the authors knowledge not available.

Several strategies to tackle the binding stoichiometry problems have been discussed in the literature. These strategies cover variation of the experimental set up or performing different or complementary experimental techniques. For example, a method called *Continuous Variation*, also known as *Job's Method*, has been very popular however critical shortcomings were identified in connection with supramolecular chemistry (*Ulatowski et al., 2016*; *Hibbert & Thordarson, 2016*). It is therefore not recommended to apply *Job's Method* for analysing binding stoichiometries. Performing additional titration experiments like UV/VIS and Fluorescence titration (*Hirose, 2006*) or analysing host-guest crystal structures with X-ray experiments (*Mazik, Cavga & Jones, 2005*; *Köhler et al., 2021*) help to understand binding stoichiometries better. However, on the one hand, X-ray experiments depend on the crystal structures which may not be obtained routinely. On the other hand,

solid state structures and solutions differ significantly, therefore a direct comparison may be less meaningful. A typical example is the structural variety of liquid water and ice (*Finney et al., 2002*; *Odelius, 2009*; *Nilsson & Pettersson, 2011*). Furthermore, additional titration experiments depend on physical properties like fluorescence activity or a measurable change of heat upon complex formation which may not always be "in place" (*Schmidtchen, 2006*; *von Krbek, Schalley & Thordarson, 2017*).

UV/VIS or Fluorescence titration can initially be analysed by determining the rank of the absorption matrix (*Wallace, 1960*). The obtained rank estimates the number of (principal) components contributing to the absorbance, giving an idea of the amount of species which have to be considered. Using the web service Sivvu.org provided by *Vander Griend et al. (2021)*, a decomposition of the absorbance matrix is available. With only a few numbers of chemical shifts tracked during NMR titration or in the limit of only one available signal, the results of the decomposition may be of limitted relevance as the number of columns in the data matrix is the upper limit of the number of factors. The same holds true in case of ITC experiments, where only the integrated heat signal is processed. Some titration curves can be analysed according to the *Molar-Ratio* or *Mole-Ratio method* (*Yoe & Jones, 1944*), where two linear functions are used to fit the experimental data. These plots can easily be generated using *SupraFit*. Although, critical shortcomings of the Mole-Ratio method have been pointed out (*Marcus, 1967*; *Momoki et al., 1969*), a brief introduction in conjunction with NMR titration will be given: The ratio of the components B and A ($\frac{B_0}{A_0}$) assigned to the intersection of the two linear functions indicates the stoichiometry. If the ratio is 1, a system with 1:1 stoichiometry was analysed. In case the ratio is below 1, 2:1 species are relevant and if the ratio is above 1, 1:2 species have to be taken into account. Vander Griend et al. (*Kazmierczak, Chew & Vander Griend, 2022c*) recently proposed an approach to overcame the first limitation (a) mentioned above. By defining the stoichiometric ratios (Eq. (3)) and the nuclearities (Eq. (4)) they reformulated the fitting problem.

$$r_i = \frac{b_i}{a_i} \qquad (3)$$

$$n_i = b_i \qquad (4)$$

Using two auxiliary functions and so called z-factors, additional to the individual stability constants the adequate stoichiometric ratios became parameters to be obtained from a hybrid nonlinear fitting process. The fitting procedure is divided in a global particle swarm optimisation to locate various minima across the stoichiometric error surface and a local optimisation to minimise the error within the global minimum. The proposed approach has been successfully tested on simulated and experimental data sets. The MatLab implementation is accessible *via* Zenodo (*Kazmierczak, Chew & Vander Griend, 2022a*) and the protocol will available at Sivvu.org.

*Hibbert & Thordarson (2016)* proposed a post-processing strategy, which includes first testing all possible models and then judging the result on the basis of scatter plots of residuals and the sum-of-squared errors (SSE) F-test. In this article, additional strategies
to support that post-processing procedure are proposed. In order to use that protocol, no additional experiments have to be performed, however the results may give rise to a modification of the experimental set up, such as considering additional ratios of $[B]_0$ and $[A]_0$ during the experiments. Analogously to the approach proposed by Thordarson, the main idea of this protocol is first to perform a statistical analysis after fitting possible models and second to compare for each used model the graphical results of this post-processing analysing. Furthermore, the resulting plots and histograms are then characterised by statistical descriptors like the standard deviation and Shannon entropy. However, the descriptors are not based on somewhat fundamental physical relations like the so-called spin contamination in the unrestricted Hartree–Fock ansatz (*Szabo & Ostlund, 1989*). Hence, no justification to interpret the absolute values is given. The concept is more based on a *rule-of-thumb* interpretation, as done for example using *fractional occupation number density* (FOD) (*Grimme & Hansen, 2015*; *Bauer, Hansen & Grimme, 2017*), which can be used to identify static electron correlation in molecules. The prior step is to estimate in which dimension the results of the descriptors are expected to be in case they indicate a model to be correct. This is realised using simulated experimental data with known stoichiometries and then fitting various models to this experimental data. Special pattern arising, if a model suits a simulated data set, help then to establish the *rule-of-thumb* approach.

Two main differences to the protocol proposed by Vander Griend et al. (*Kazmierczak, Chew & Vander Griend, 2022c*) have to be mentioned: In the approach of Vander Griend et al. flexible stoichiometries are taken into account, while this approach was tested with binding models having species of 1:1, 2:1 and 1:2 stoichiometries. These models were the first ones implemented during the development of SupraFit as they are the most common models which are considered in the recognition of carbohydrates with artificial receptors (*Mazik, 2009*; *Mazik, 2012*; *Miron & Petitjean, 2015*; *Amrhein, Lippe & Mazik, 2016*; *Kaiser, Geffert & Mazik, 2019*; *Francesconi & Roelens, 2019*; *Davis, 2020*).[1] The second difference compared to the protocol according to Vander Griend et al. is the evaluation of the models. This proposed approach is based on the a posteriori evaluation of the models using various statistical approaches while Vander Griend et al. use the root-mean-square error after the optimisation of the parameters to evaluate the goodness of a model.

In this article, three different protocols are introduced which can be applied in order to analyse the stoichiometries of species relevant in NMR titration experiments. Furthermore, the proposed protocol may be independent of NMR titration experiments, but an application to other curve-fitting problems has to be studied in detail, which is out of the scope of this article. All proposed methods are implemented in the recently presented program *SupraFit* (stable version 2) (*Hübler, 2019*; *Hübler, 2022b*) and are accessible through an intuitive user interface. Exploiting the simulation functions in *SupraFit*, the protocol can be tested for own problems. This article is organised as follows. An introduction to Monte Carlo simulation and the statistical descriptors as well as Cross Validation and the new protocol termed Reduction Analysis are given in 'Methods and Implementation'. The main aspects of the implementation in *SupraFit* are given as well. After a short explanation of the data generation in 'Data Generation', the results

[1] Clearly, binding models having more complex stoichiometries were also used, however reported less frequently, which may be due to either the limitation of the used programs or the lack of justification to use a more complex model. However, in the recent development version of SupraFit, binding models with any stoichiometry $A_aB_b$ can be applied in conjunction with NMR and UV/VIS titration as well as ITC experiments.

are discussed in 'Results'. In that context, four simulated NMR titrations of different stoichiometries are each analysed using the available NMR titration **models** with a post-fitting analysis using the statistical methods. The final statistical descriptors are calculated from these results and discussed in detailed. The final aspects are summarised in 'Summary and Conclusion'. In that article simulated data sets are highlighted with underlines to make them easily distinguishable from the **models** fitted, which are highlighted using bold text.

## METHODS AND IMPLEMENTATION

### Monte Carlo simulation (MC)

The usage of Monte Carlo simulation (MC) and Bootstrapping (BS) to estimate confidence intervals using the percentile methods as proposed previously (*Thordarson, 2011*; *Lowe, Pfeffer & Thordarson, 2012*) has already been presented in an earlier article (*Hübler, 2022b*). Recently, *Kazmierczak, Chew & Vander Griend (2022b)* have analysed BS in context of photometric titration experiments, including stock solution errors.

After performing MC or BS, the confidence interval does not contain information of the distribution of individual obtained values for the parameters as only the percentile is calculated (*Efron, 1979*). The dimension of the distribution *e.g.*, the number of Monte Carlo runs and the individual probability of any value to be obtained after a single run are not taken into account. To include the number of Monte Carlo steps and the probability of an individual parameter value, the histograms are further characterised using the standard deviation (Eq. (5)) and the Shannon entropy (Eq. (6)).

$$\sigma_{dist} = \sqrt{\frac{\sum_{i}^{N}(x_i - \overline{x})^2}{N-1}} \tag{5}$$

$$H(x) = -\int p(x_i) ld(p(x_i)) \tag{6}$$

While the standard deviation is a common tool in every-days statistical judgement, the Shannon entropy is more often used in pattern recognition and information theory (*Bishop & Nasrabadi, 2006*). It is out of the scope of this article and *SupraFit* to deal with the various aspects in pattern recognition, but the most important facts shall be summarised. If only one possible result for a parameter is obtained, the probability $p(x)$ is 1, resulting in an entropy of 0. If however the possible results for a parameter are uniformly distributed across $N$ possible values, the entropy is maximum with $-ln(1/N)$. As consequence, the lower values of the Shannon entropy indicate sharper peaks, while higher entropy indicate broader peaks. In case of a normal distribution, the standard deviation and the Shannon entropy are directly linked by Eq. (7) and the Shannon entropy gives no additional information. However, for non-normally distributed random numbers (*Bishop & Nasrabadi, 2006*) like in case of the distribution of the parameter values after Monte Carlo simulation in connection with nonlinear regression, Eq. (7) is not true and the Shannon entropy accounts

for the distribution of the random numbers.

$$H(x) = \frac{1}{2}(1 + ln(2\pi\sigma^2)) \tag{7}$$

As the Shannon entropy does not depend on the concrete values of the parameters $(x)$ itself but on the probability $(px(x))$, the Shannon entropy is dimensionless. The standard deviation however has the dimension of the value. The comparison of the results of Monte Carlo simulation with parameters of different dimensions is hence only meaningful if the dimensionless Shannon entropy is used.

The discrete calculation of the Shannon entropy is given in Eq. (8), where $B$ denotes the number of bins and $\Delta$ the width of a single bin.

$$H(x) = -\sum_{i}^{B} p(x_i)\Delta ld(p(x_i)) - ld\Delta \tag{8}$$

Since the Shannon entropy is in *SupraFit* calculated with the same number of bins for each distribution individually, the width of a single bin depends on the distribution of each parameter itself. For more compactly distributed values of a parameter, smaller bins are obtained resulting in artificially increased entropy values. Hence the second term in equation is omitted in the standard calculation and the Shannon entropy is calculated according to Eq. (9) if not requested otherwise.[2]

$$H(x) = -\sum_{i}^{B} p(x_i)\Delta ld(p(x_i)) \tag{9}$$

### Cross Validation (CV)

Cross Validation defines a resampling method (*Efron, 1979*), which usually is applied to determine the ideal parameters or latent variables in multivariate statistics such as PCA or PLS for example in QSAR studies (*Gramatica, 2007*). The parameters or latent variables are judged according their predictive behaviour, which can in the simplest case be determined as follows: Having a data set with N elements, in Leave-One-Out Cross Validation (L1O-CV) N-1 data points are used to train a new model using a set of chosen parameters or number of latent variables. The performance of the newly defined model can be evaluated using the omitted data point which forms the test set. As there are N independent possibilities to form a training set, N independent evaluations can be used to judge the performance of the variables. Analogously, in Leave-Two-Out Cross Validation (L2O-CV) pairs of two points configure the test set. The training set is composed of the remaining N-2 data points, with N*(N-1) possible combinations. As the order is not relevant, only N*(N-1)/2 combinations are unique.

In *SupraFit*, CV is not used in the described manner, which is to recalculate the test set data points and estimate the performance of the variables on the base of statistical judgment. Instead, it is implemented to judge the models if hypothetically fewer data points were acquired. In the simplest approach (L1O-CV), each data point of the experimental data set is left out once and the model is fitted. In the limit of only one tracked NMR signal or

[2] In the current development version of *SupraFit* this can be requested in the settings.

[3]If the maps were calculated individually within separate threads, the threads or maps had to be synchronised before the fitting started. Otherwise the uniqueness of the sequences is not ensured.

[4]Having a data set with 20 spectra and leaving hypothetical 10 data points out, S is $20!/2 \cdot 10! = 184756$, which can be realised within a few minutes on a modern eight core systems with hyper-threading enabled. On the other hand, $X = 5$ and $N = 100$ result in $S_{max} = 7.5 \cdot 10^7$, which might take several GB of RAM to simply store the map, of which for example only $S = 10^4$ replications are used at the end.

an ITC experiment, only single points can be left out. Having more than one NMR signal analysed or several wavelength in case of photometric experiments, each point on the x axis is assigned a vector of observed values, which will be simultaneously left out. However, they will still be referred as data points within this context. As there are again N ways to leave one point out, a distribution of N model parameters can be obtained. The analysis of the distribution is performed in the same fashion as introduced in case of the Monte Carlo simulation. Furthermore, L2O-CV as well as the generalisation Leave-X-Out Cross Validation (with X < N) can be performed and analysed within *SupraFit*. In case of the Leave-X-Out Cross Validation, the number of trials can be calculated using the general formula for combinations (Eq. (10)).

$$S_{max} = \frac{N!}{X!(N-X)!} \tag{10}$$

As advantage over the Monte Carlo Simulation, where an additional input parameter is required and the results are not exactly reproducible, Cross Validation works without the additional input and leads to reproducible results for the same combinations.

The individual parameter fitting procedures during Cross Validation are performed as if there were Monte Carlo simulation, including the parallelisation. A benchmark on the MC implementation with respect to the parallelisation was given previously (*Hübler, 2022b*). However some aspects of the data preparation, that is (a) the generation of the map containing the pseudo-experimental data with left out points and (b) the selection of the pseudo-experimental data, will be discussed in this context. In general, the valid combination of points to be left out will be precalculated in order to maintain unique calculation during parallel execution.[3] In case of L1O and L2O, the number of parameter fitting processes are N and N*(N-1)/2. Both can easily generated using a simple loop or two nested loops, and if $N < 100$ routinely be evaluated in *SupraFit*. In case of more points X to be left out and/or significantly larger amount of data points N acquired, the number of possible combinations $S_{max}$ easily reaches the limit of computational time and resources.[4] As consequence a random subset of the combinations has to be chosen prior to the parameter fitting. In case of a greater number of N and the chosen value of X, the preparation of all combinations $S_{max}$ may take longer than the fitting of the subset with S combinations itself. Therefore two approaches for the generation of the map are implemented in *SupraFit*. (a) The whole map is calculated using nested loops. In case there are fewer steps S than possible steps $S_{max}$ requested, the individual pseudo-experimental data sets are randomly chosen from the fully generated map using uniformly distributed random numbers. (b) In the second approach, uniformly distributed random numbers are used to generate S sequences (having the length X) of unique numbers which define the points to be left out. Uniqueness of the sequences itself is ensured after sorting a newly obtained one first and then comparing it to the previously generated sequences. A new unique number within the sequences becomes more improbable as the sequence gets more complete ($1/X$ for the last number) and the uniqueness of the sequences becomes more improbable as the number of already stored sequences increases. Therefore, this approach becomes only more efficient compared to the nested loop precalculation if significantly

fewer steps $S$ than $S_{max}$ are requested. Depending on the number $S$ and the ratio of $S$ and $S_{max}$, *SupraFit* choses one approach in favour of the other, but each of the two algorithm can be enforced from the user interface.

### Reduction Analysis (RA)

During experimental design, the number of data points to be acquired and the ratio of the initial concentration of the substances B and A to be analysed is of particular interest. It has already been discussed by Thordarson, that for many systems ten points between a ratio $\frac{B_0}{A_0}$ of 0 and 1.5 are most important, and additionally ten more up to a ratio of 50 should be acquired (*Thordarson, 2011*). *SupraFit* provides a new protocol called Reduction Analysis (RA) which exploits possible redundancy if data points are included in the data set, which itself do not change the model parameters. The fitted parameter of a 1:1 model on a hypothetical experiment with 1:1 stoichiometry would not differ if fewer data points where included during the fitting process, as long as more data points as essentially necessary for an adequate determination for the 1:1 species are available. This approach is comparable to the *Mole-Ratio method*: After the saturation point($\frac{B_0}{A_0} > 1$) is reached, the observed chemical shift changes less than in the range of ($0 < \frac{B_0}{A_0} < 1$). Limitation itself may arises if the stability constants are low and saturation of the complex formation is not reached within that threshold ratio, that is defined by the stoichiometry of the complexes. While the condition of sufficiently stable complexes can not be expected a priory, the potential of Reduction Analysis to indicate if model parameters are not appropriate will be discussed within this article.

The automatic procedure is based on the step-wise reduction of the number of data points beginning at the end. After each removal, the model is fitted to the remaining points of the data set. The best-fit parameters $\hat{\theta}$ are stored and finally plotted as "function" of the highest ratio, which is the ratio of the last available data point. Since titration should reach at least the saturation point, all fitted parameter below the particular ratio are assumed to differ significantly, even if the stoichiometric model is correct. To allow a rational comparison of Reduction Analysis performed on different models, the standard deviation of the parameters are calculated. Since every model needs a different ratio to be reached during titration, the comparison of the naive standard deviation is not meaningful. Instead the *partial standard deviation* $\sigma_{pt}$ (Eq. (11)) is calculated, taken all data points above a cut-off ratio into account. This cut-off is defined by the "highest necessary" saturation point, therefore ensuring, that for all models the saturation point has been reached. In *SupraFit* this cut-off is automatically set for NMR titration experiments according to the tested models but may be changed to any value if necessary.

$$\sigma_{pt} = \sqrt{\frac{\sum_{i=k}^{N}(\hat{\theta}_i - \bar{\hat{\theta}})^2}{N - k - 1}} \tag{11}$$

with: N - last data point

k - first data point

## DATA GENERATION

To introduce the Monte Carlo approach and the resampling plans, the same simulated NMR titration as used previously (*Hübler, 2022b*) was taken. The concentration of component A was set to $[A]_0 = 10^{-3}$ mol/L and component B was added up to a ratio $\frac{B_0}{A_0}$ of 3.98. Additionally simulated data for a plain 1:1 model as well as of models with mixed 2:1/1:1 and 2:1/1:1/1:2 stoichiometries were used. The stability constants (*lgK*) were chosen as random numbers between 1 and 5. The upper limit of *lgK* = 5 in case of NMR titration was discussed by Thordarson previously (*Thordarson, 2011*). In photometric titration experiments, the sensing limit of the stability constants using global analysis was identified to be $K[A]_0 < 1000$, which is satisfied by the chosen conditions (*Kazmierczak et al., 2019*). On top of the ideal titration curve of seven tracked NMR signals in a range between 0 and 8 ppm, random numbers with $\sigma_{MC} = 0.001$ were added to the chemical shifts to account for experimental noise; errors in stock concentrations were not included explicitly. The simulated data sets are included in the Zenodo archive (*Hübler, 2022a*). Each simulated data set was analysed with the four available models as follows: The best-fit parameters were obtained from nonlinear regression and then post-processed using Monte Carlo simulation with 2,000 and 10,000 steps, Cross Validation ($X = 1, 2, 3, 4$ and 5) and Reduction Analysis. In case of the L5O-CV, the default of 10,000 individual replications were randomly chosen from the 15,505 possible combinations. The results of the statistical post-processing were then used to estimate standard deviation and Shannon entropy values characterising the histograms and the partial standard deviation in case of the Reduction Analysis. Simulated data were preferred over experimental data to ensure that the underlying model is well defined. As a result, the various fitted models and therefore the statistical descriptors can then be judged according their performance to identify the original model.

## RESULTS

### Data set with 1:1/1:2 stoichiometry
#### *Monte Carlo Simulation*

Exemplary results of the Monte Carlo simulation (with $S = 2000$, $\sigma_{\mathrm{mc}} = SE_y$) of all four models on top of the 1:1/1:2 data set are given in Figs. 1 and 2. The plots show both, all parameters and stability constants only. The factor analysis of the chemical shifts using Sivvu.org reveals three factors (Fig. S20B), indicating that three species contribute to observed signals. This is in agreement with the original model, however no further information about the stoichiometry of the species can be deduced using the analysis.

Following the "visual inspection," the comparison of the histograms already gives an estimation of the quality of the used models. Since the **1:1/1:2 model** recovers the original parameters, the results of the Monte Carlo simulations on top of the best-fit parameters of the corresponding **1:1/1:2 model** are considered as ideal results, having appropriate distributions of the parameters and therefore standard deviation and Shannon entropy values indicating well estimated model parameters. The histogram of each parameter shows a narrow distribution (Fig. 1), which is also true for the simpler **1:1 model**. The MC simulation results of the **2:1/1:1 model** give rather broad distributions (Fig. 2a), and since
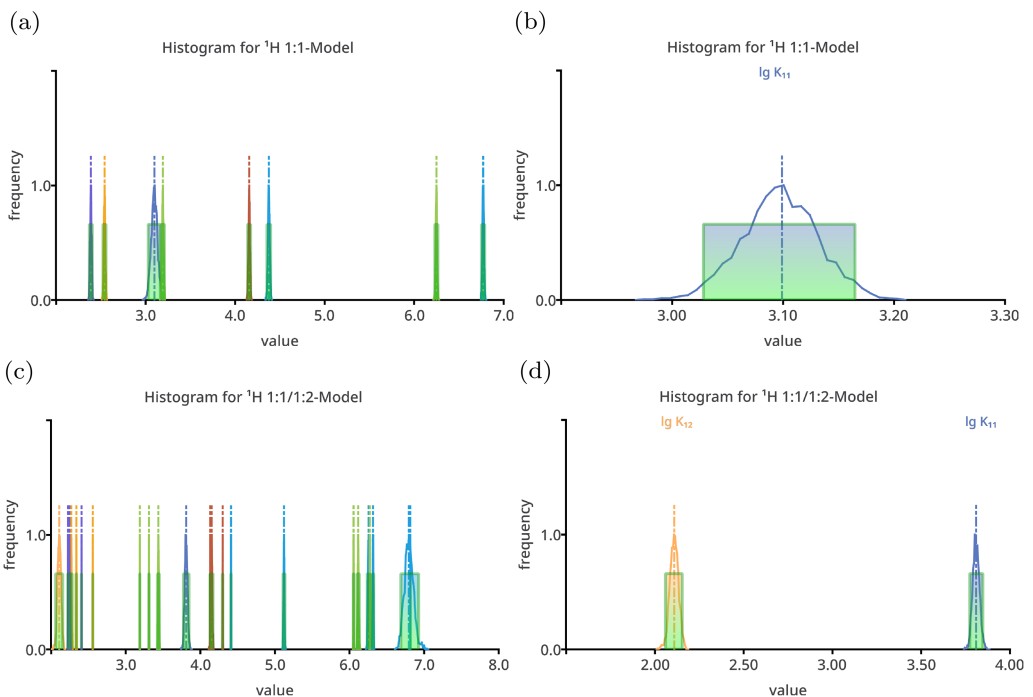

**Figure 1** **Histogram like charts after performing Monte Carlo simulation on top of the best-fit 1:1 model (A, B) and the best-fit 1:1/1:2 model (C, D) fitted to the simulated titration curve with 1:1/1:2 stoichiometry .** The charts in (A) and (C) show all model parameters, the dimensionless stability constants as log10 and chemical shifts in ppm, which is the standard way to plot Monte Carlo simulation histograms in *SupraFit*. The charts (B) and (D) show only the stability constant, with the corresponding parameter name above the curve. Apart form the incorrectly estimated value of lg $K_{11}$ in the 1:1 model, the distribution of the "observed" stability constants and chemical shifts are narrow and do not indicate an inappropriate model.

all parameter are plotted at once, they overlap. Neglecting chemical shifts and visualising only the stability constants, the histograms reveal a broader distribution of the lg $K_{21}$ and a narrower distribution of the lg $K_{11}$ values (Fig. 2B). The values of the stability constant remain physical meaningful between one and three. The histogram of the most complex model show only broadly distributed parameters, and focusing on the stability constants only, lg $K_{21}$ ranges from $-7.5$ to $3.0$ being clearly non-physical. The correct parameter appear to be narrowly distributed. Hence, from pure "visual inspection" the **1:1/1:2 model** and **1:1 model** with both having narrower distributed parameters appear to behave much better than the **2:1/1:1** and **2:1/1:1/1:2 model** with broader distributed parameters.

However, the "visual inspection" is rather biased and need strictly comparable charts (scaling, resolution, *etc.*). A more unbiased way utilises statistical descriptors derived from the distribution of the parameters itself. In case of the stability constants, these statistical descriptors—the confidence intervals and the values for the Shannon entropy and standard deviation—are given for all four tested models in Table 1. As stated above, the results from the **1:1/1:2 model** are taken as reference. The results of the remaining models will then be compared to the reference result.

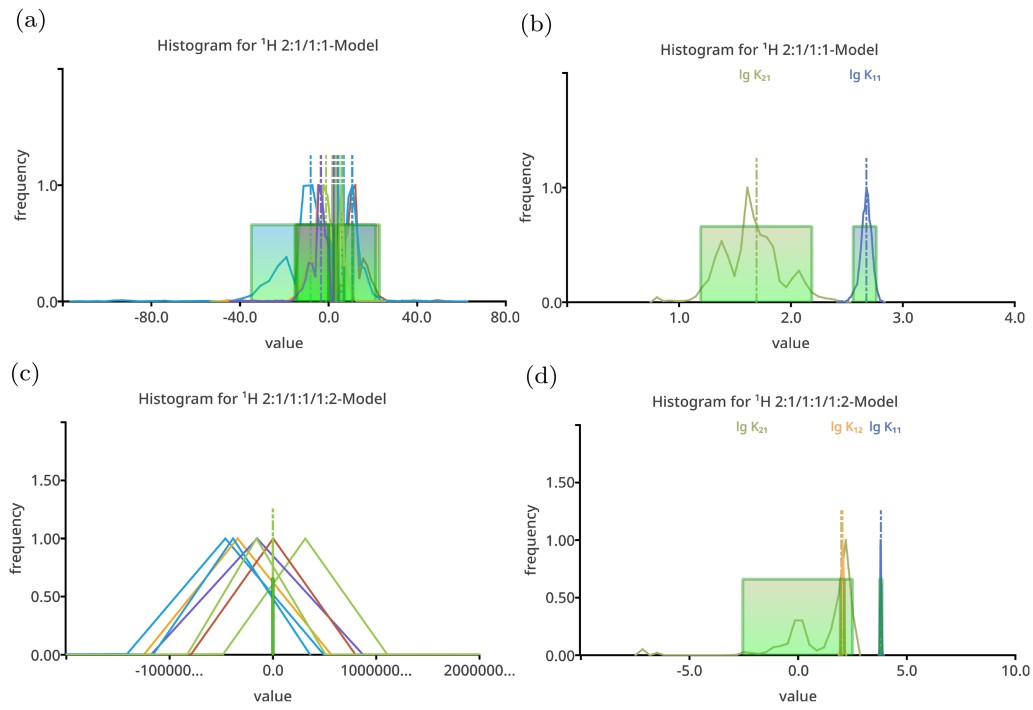

**Figure 2** **Histogram like charts after performing Monte Carlo simulation on top of the best-fit 2:1/1:1 model (A, B) and the best-fit 2:1/1:1/1:2 model (C, D) fitted to the simulated titration curve with 1:1/1:2 stoichiometry.** The charts (A) and (C) show all model parameters, the dimensionless stability constants as log10 and chemical shifts [ppm], while the charts (B) and (D) show only the stability constants. The distribution of the individual parameters—especially the distribution of the chemical shifts—is broad. Hiding the chemical shifts, the distribution of the appropriate stability constants is revealed to be comparable narrow, in particular the distribution of lg $K_{11}$ in the **2:1/1:1 model**. The distribution of the lg $K_{11}$ gets worse changing the **2:1/1:1 model** to the more complex **2:1/1:1/1:2 model**.

**Table 1** **Stability constants of the four tested models, the 95% confidence intervals obtained from Monte Carlo simulations including the median, $\sigma_{dist}$ and $H(x)$ calculated from the distribution of the parameters, partitioned into 30 bins.**

| parameter | $\hat{\theta}$ | $\Delta\theta+$ $\Delta\theta-$ | $[\theta_-$ | $\theta_+]$ | $\sigma_{dist}$ | $H(x)$ |
|---|---|---|---|---|---|---|
| | | 1:1/1:2 model; SSE = 0.0001; $SE_y$ = 0.0011 | | | | |
| $lgK_{11}$ | 3.81 | +0.04 −0.04 | 3.77 | 3.85 | 0.0188 | 0.0523 |
| $lgK_{12}$ | 2.11 | +0.04 −0.05 | 2.06 | 2.15 | 0.0242 | 0.0640 |
| | | 1:1 model; SSE = 0.0365; $SE_y$ = 0.0171 | | | | |
| $lgK_{11}$ | 3.10 | +0.07 −0.07 | 3.03 | 3.17 | 0.0344 | 0.0837 |
| | | 2:1/1:1/1:2 model; SSE = 0.0001; $SE_y$ = 0.0011 | | | | |
| $lgK_{21}$ | 1.99 | +0.51 −4.53 | −2.54 | 2.50 | 1.6933 | 1.4813 |
| $lgK_{11}$ | 3.81 | +0.06 −0.06 | 3.75 | 3.87 | 0.0294 | 0.0796 |
| $lgK_{12}$ | 2.04 | +0.09 −0.10 | 1.94 | 2.14 | 0.0496 | 0.1164 |
| | | 2:1/1:1 model; SSE = 0.0018; $SE_y$ = 0.0039 | | | | |
| $lgK_{21}$ | 1.69 | +0.49 −0.50 | 1.18 | 2.18 | 0.2637 | 0.4507 |
| $lgK_{11}$ | 2.67 | +0.09 −0.12 | 2.55 | 2.76 | 0.0505 | 0.1336 |

All parameters are unsymmetrically distributed around the best-fit values, which is in agreement with the general findings by Motulsky and Christopoulos (*Motulsky & Christopoulos, 2003*). A detailed analysis in case of stability constants has later been done by Thordarson (*Thordarson, 2011*) and Vander Griend (*Kazmierczak, Chew & Vander Griend, 2022b*). In the **1:1 model**, the confidence limits for lg $K_{11}$ are a nearly twice the limits of lg $K_{11}$ in the **1:1/1:2 model**. Furthermore, the standard deviation and Shannon entropy for lg $K_{11}$ are higher in the pure 1:1 model. According to the descriptors, the correct parameters in the 2:1/1:1/1:2 model behave slightly worse than in the 1:1/1:2 model, but the confidence interval as well as $\sigma_{dist}$ and H(x) are in the same order of magnitude. On the other hand, for the parameter lg $K_{21}$ the simulation lead to the widest confidence interval and the worst values of $\sigma_{dist}$ and H(x). The parameter lg $K_{11}$ in the **2:1/1:1 model** is connected to a value of the Shannon entropy between one obtained in case of the pure **1:1** and the mixed **2:1/1:1/1:2 model**, but with simultaneously having the worst values of $\sigma_{dist}$ and largest confidence interval. In contrast to lg $K_{21}$ in the **2:1/1:1/1:2 model**, the statistical outcome in the **2:1/1:1 model** is much better: a tighter confidence interval and smaller values of $\sigma_{dist}$ and H(x) were obtained. As lg $K_{11}$ and lg $K_{12}$ are sufficient to recover the simulated titration curve, lg $K_{21}$ can freely accept any value with only slightly improving the resulting *SSE*. If the parameters linked to the chemical shift were not included in charts, from pure inspection of the stability constant without comparing to other models, the **2:1/1:1 model** could still be acceptable, with the unknown source of broader confidence interval maybe rooted in experimental circumstances. Taking all parameters into account (Figs. 2A and 2C), it is without comparing to other models clear, that the **2:1/1:1** and the **2:1/1:1/1:2 model** are unsuited for the data set since some of the individual parameter take even negative values. On the other hand using this judgment, the **1:1 model** describes the data sufficiently well.

Figure 3 shows the Shannon entropy (H(x)) and the standard deviation ($\sigma_{dist}$) of the Monte Carlo simulations with 2000 steps compared to simulations with 10000 steps. The qualitative behaviour of both descriptors is identical for the tested models: The lowest obtained values are assigned to the parameters of the **1:1/1:2 model**. According to the individual values of descriptors characterising the stability constants, lg $K_{21}$ is inappropriate while lg $K_{11}$ and lg $K_{12}$ are adequate to obtained well fitted models.

The former analysis indicates, that in order to ensure the qualitative behaviour of the statistical parameters, the number of steps $S$ in Monte Carlo simulations is of less importance. However, for each model the according $\sigma_{MC}$ value obtained after fitting was used, which is the default input for the Monte Carlo simulation. Hence the different outcome of the Monte Carlo simulation could be traced back to the different $\sigma_{MC}$ values. In a subsequent analysis, the input standard deviation $\sigma_{MC}$ was then varied as follows: Monte Carlo simulations of each best-fit model were repeated with all possible $SE_y$ value obtained from the best-fit parameters of the four models. The results, visualised as bar charts in Fig. 4, show the dependency of the Shannon entropy from the input standard deviation for each model. In Fig. 4, results depicted with the letter A were obtained using $SE_y$ from the **1:1 model** ($SE_y = 0.01708$) and with the letter B $SE_y$ from the **2:1/1:1 model**

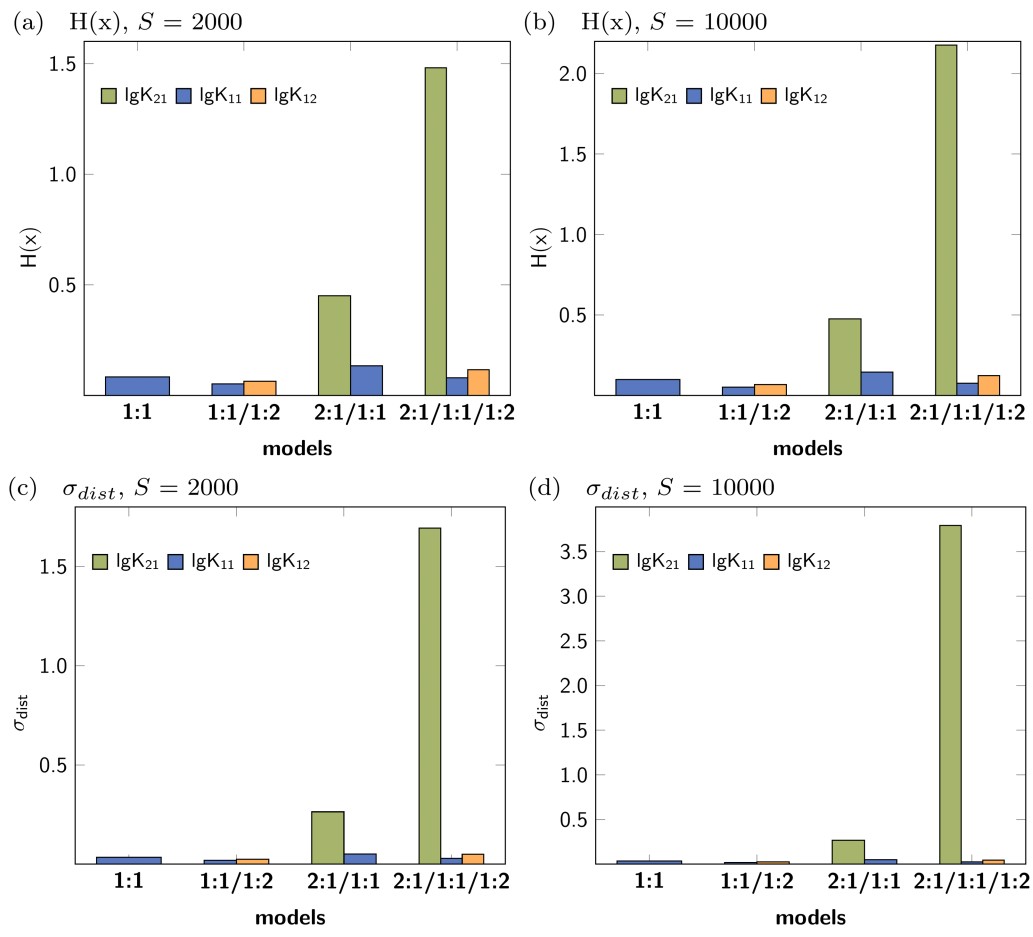

**Figure 3  Calculated Shannon entropy (H(x)) and standard deviation ($\sigma_{dist}$) of the individual stability constants after Monte Carlo simulations with $S$ steps.** The lowest values were obtained for the 1:1/1:2 model.

($SE_y = 0.003889$) was used. The results depicted with C are linked to the **1:1/1:2 model** ($SE_y = 0.00106$) and with D to the **2:1/1:1/1:2 model** ($SE_y = 0.00108$).

The **1:1 model** with the fewest parameters exhibits the worst fit, which is indicated by the highest value of SSE and $SE_y$ (compare Table 1). Hence the individual values of the descriptors obtained from Monte Carlo simulation with input A indicate worse performance than in case of the input B to D, which are connected to lower values of $SE_y$. The bar charts in Fig. 4 show clearly lower values of H(x) in case the input $SE_y$ was taken from a more complex model. Furthermore, the entropy falls below the values obtained using correct 1:1/1:2 model when the lower $SE_y$ from the 2:1/1:1/1:2 model is applied. The $SE_y$ value from the **1:1/1:2 model** (C) is the second-best result, therefore using the $SE_y$ value obtained from the fit of the **1:1 model** (A) or **2:1/1:1 model** (B) worsen the results of the Monte Carlo simulations clearly. As the $SE_y$ obtained from the **2:1/1:1/1:2 model** (D) is the lowest, the best $\sigma_{dist}$ and $H(x)$ results are obtained using this input. In the same manner, the resulting entropy gets lower when the Monte Carlo simulations of the **2:1/1:1**

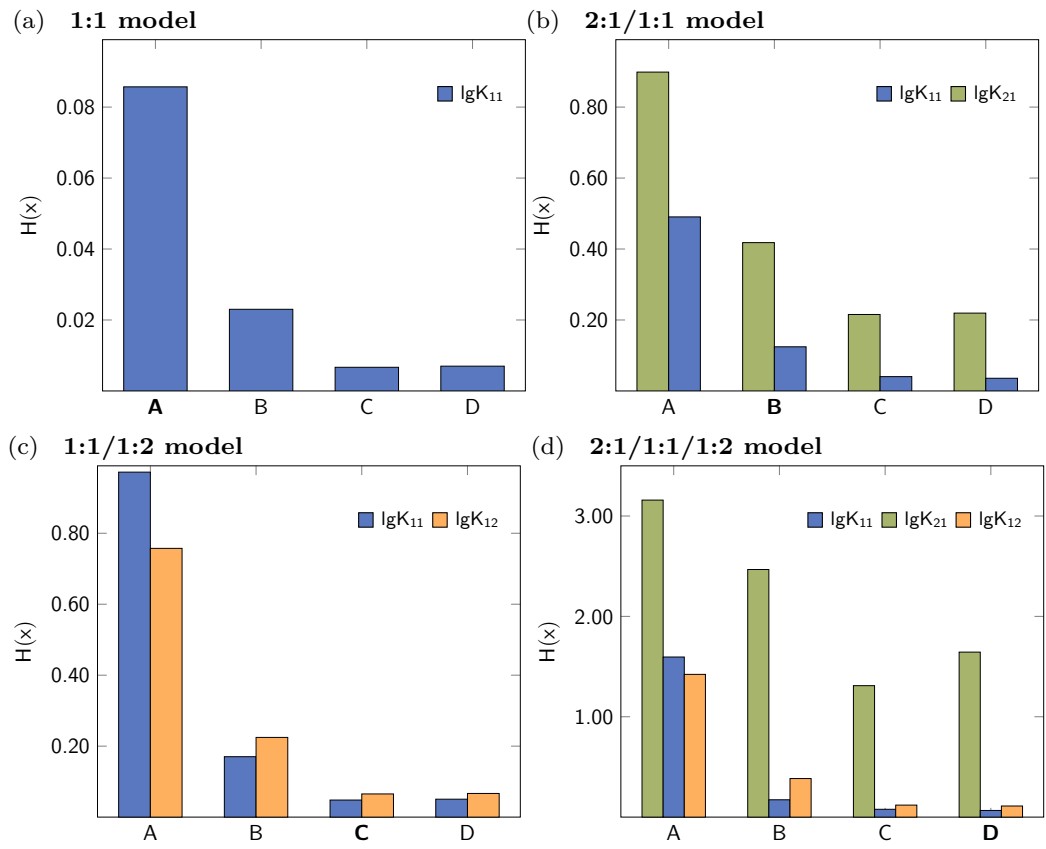

**Figure 4** **Comparison of different Shannon entropy values.** Each bar in the charts corresponds a Monte Carlo simulation with $\sigma_{MC}$ according to the assigned letter. The letter is typed bold if $\sigma_{MC} = SE_y$ coming from the original model. The $SE_y$ input for the calculations are (A) $SE_y = 0.01708$, (B) $SE_y = 0.00388$, (C) $SE_y = 0.00106$, (D) $SE_y = 0.00108$.

**model** are performed with $SE_y$ input from **1:1/1:2** or **2:1/1:1/1:2 model** (Fig. 4B). On the other hand, the Shannon entropy increases if the $SE_y$ value from the **1:1 model** is applied. Common for all calculations is the bad performance of lg $K_{21}$ in all models regardless of the input $SE_y$. On the other hand, the entropy values of the correct parameters lg $K_{11}$ and lg $K_{12}$ decrease systematically upon lowering the $SE_y$.

As alternative to the input of a standard deviation to simulate experimental error, Bootstrapping (BS) can be applied. During BS the residuals after fitting are randomly added to the best-fit titration curve to recover the original error. The Shannon entropy for each stability constants after applying Bootstrapping ($S = 2000$) to each model are visualised in Fig. 5. As observed previously, the Shannon entropy indicates lg $K_{21}$ as inappropriate stability constant and furthermore the **1:1/1:2 model** as the best fitting model.

The application of Monte Carlo simulation after obtaining the best-fit model shows that not only the most probable model should be tested - more information arise from explicit wrong and overfitted models, as in the **2:1/1:1/1:2** sample. The correct model parameter show lower H(x) and $\sigma_{dist}$ values, while the incorrect parameters in the complex model are

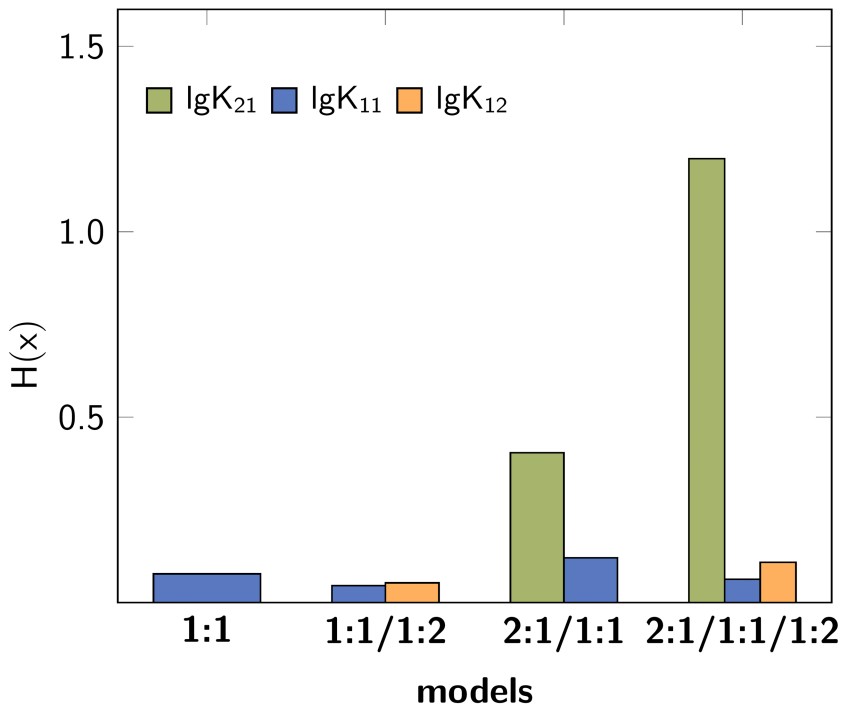

**Figure 5** Calculated Shannon entropy after Bootstrapping with $S = 2000$.

linked to higher values of the descriptors. On the other hand, Monte Carlo Simulation did not indicate that the **1:1 model** is not appropriate.

### Cross validation

Leave-X-Out Cross Validation (LXO-CV) analysis with $X = 1 - 5$ was performed and the Shannon entropy and the standard deviation of the histograms obtained for each stability constants were calculated. The Shannon entropy for each LXO-CV run for all stability constants are given in Fig. 6. Individual values obtained per model and parameter follow the trend already obtained in the Monte Carlo simulation. The entropy for the correct parameters (lg $K_{11}$ and lg $K_{12}$) tend to be lower than in case of the incorrect stability constant lg $K_{21}$. Furthermore lg $K_{11}$ is lowest in the **1:1/1:2 model**, followed by the 1:1 model. All this results are obtained with no supplementary input given in contrast to the Monte Carlo Simulation, where a $SE_y$ has to be defined.

The concrete values for H(x) indicate that upon using higher order Cross Validation—leaving more data points out—the entropy and the standard deviation increase. Noteworthy, regardless of the type of the Cross Validation, the statistical descriptors obtained for the model parameters indicate best performance for both lg $K_{11}$ and lg $K_{12}$.

### Reduction analysis

Reduction analysis was performed on all tested models with the results shown in Fig. 7. Upon continuously removing the very last data point from the data sets in case of the **1:1 model** (Fig. 7A), lg $K_{11}$ constantly increases until a ratio of $\frac{[B]_0}{[A]_0} = 1.8$. The best-fit value

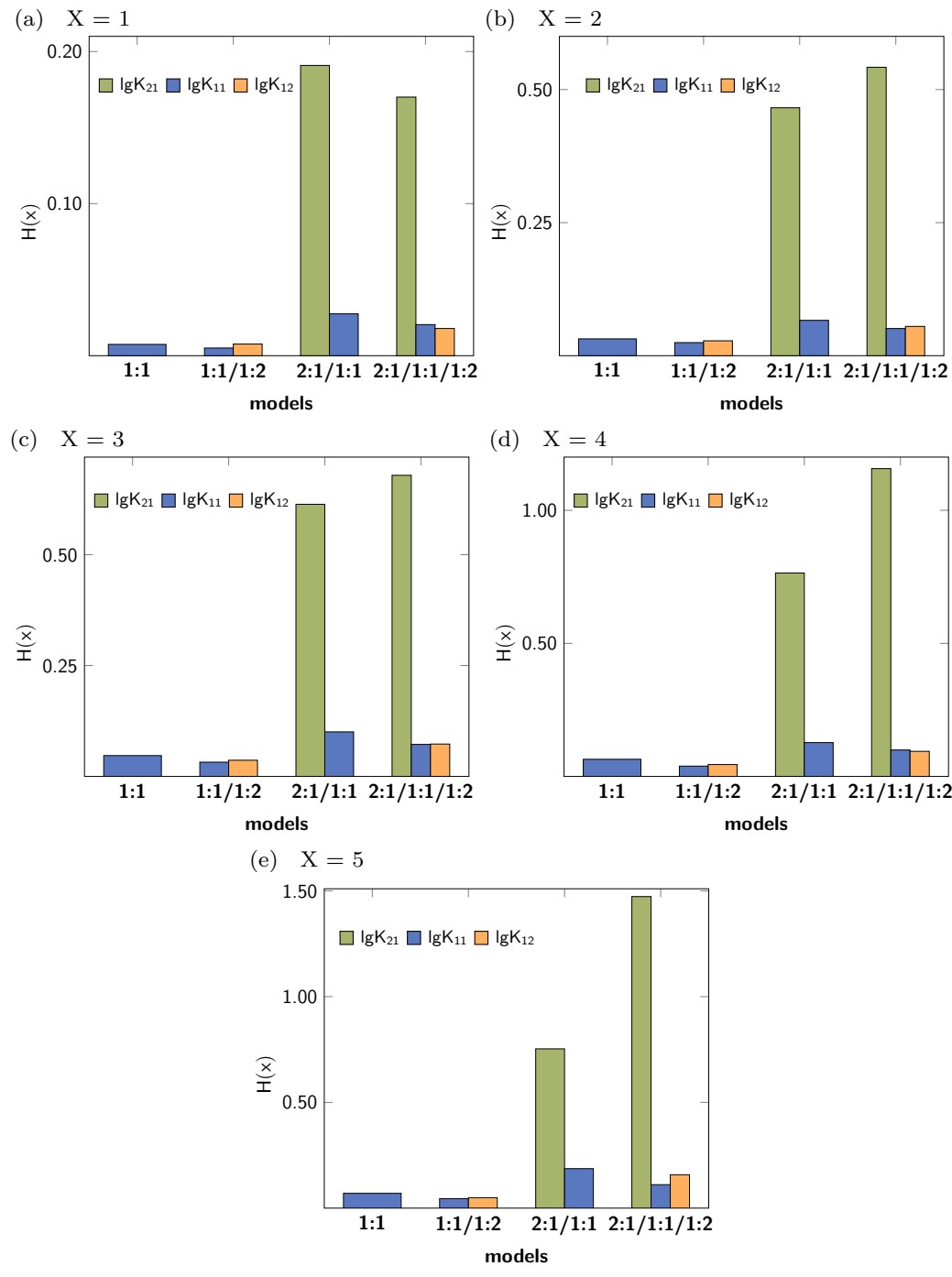

**Figure 6** (A–E) Shannon entropy of the histograms obtained for the distribution of the individual stability constants after several Cross Validation calculation for each model.

of lg $K_{11}$ goes up to 3.2. Afterwards, having only data points below a ratio of 1.8 included, the best-fit value of the stability constants decreases. In contrast to that, the individual parameters in the **1:1/1:2 model** do not show such trend above a $\frac{[B]_0}{[A]_0}$ ratio of 2.0. However,

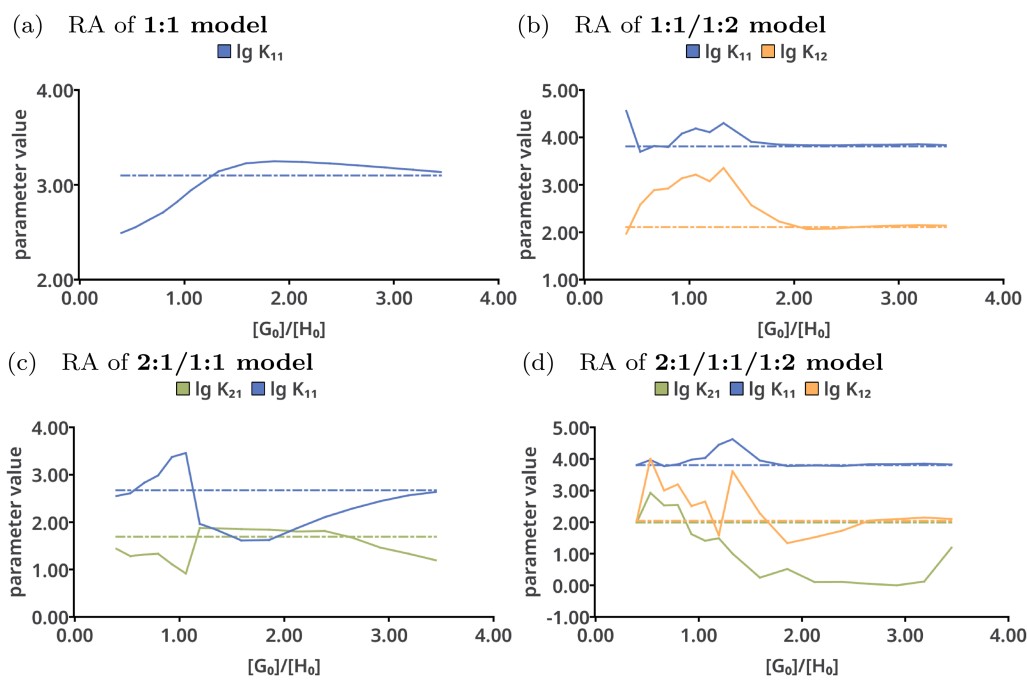

**Figure 7** **Results of the Reduction Analysis for each tested model.** The solid lines indicate the optimised parameters if the titration was only performed up to the ratio according to the current x value. The dotted lines indicate the optimised parameters after complete titration. In the chart (A) (**1:1 model**), the deviation from the fitted value upon removing the last data point can be observed. Although scaled differently, the optimised parameters of the **1:1/1:2 model** remain constant above a ratio of 2.0 (B). The analysis shown in (C) and (D) with wrong models indicate more deviation of the refitted parameter compared to the originally optimised one, except for lg $K_{11}$ in the **2:1/1:1/1:2 model**, which behaves similar to lg $K_{11}$ in the **1:1/1:2 model**.

using only data points below that ratio, the fitted parameters differ from the originally fitted values. According to that analysis, for correct description of a **1:1/1:2 model** at least a ratio of 2.0 is mandatory, which is in line with previous rule-of-thumbs for the *Mole-Ratio method*. In contrast to the **1:1/1:2 model**, the trend in the stability constants of the **2:1/1:1 model** show more deviations from the best-fit value upon removing the last data points. The values for lg $K_{11}$ in the **2:1/1:1/1:2** remain nearly the same above a ratio of 2.0, but differ more below that ratio.

The rationalisation of Reduction Analysis is accomplished using the above introduced partial standard deviation $\sigma_{pt}$ (Eq. (11)) where different cut-off values were applied. As cut-off values both a ratio of 1.8 and 2.0 were chosen. The results are represented as bar charts in Fig. 8. Although the values of $\sigma_{pt}$ allow rational comparison of reduction analysis, empiricism is mandatory. As the **1:1/1:2 model** should recover the simulated data best, the resulting values for $\sigma_{pt}$ are again considered as ideal or reference results. Starting with lg $K_{11}$ and a cut-off ratio of 2.0, the corresponding $\sigma_{pt}$ decreases from 0.04037 in the 1:1 model to 0.00724 in the **1:1/1:2 model**. In the **2:1/1:1/1:2 model**, $\sigma_{pt}$ reaches 0.02700 in case of lg $K_{11}$. In the **2:1/1:1 model**, the partial standard deviation for lg $K_{11}$ exhibits to highest value (0.29115). In analogy, $\sigma_{pt}$ for lg $K_{12}$ increases from 0.03485 about a factor of seven changing

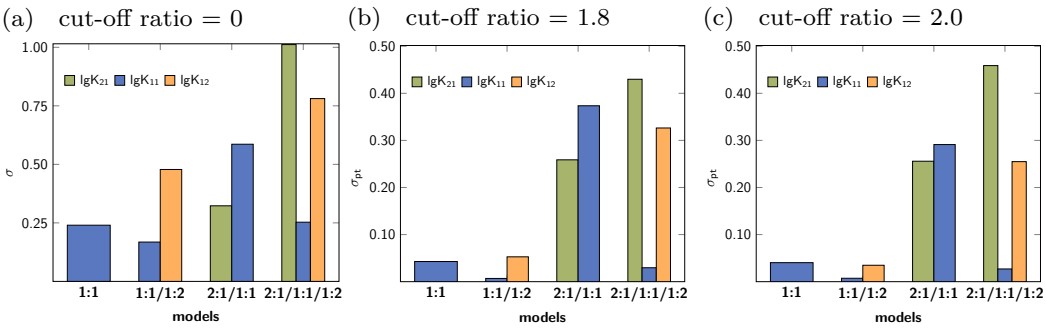

**Figure 8** **(A–C) The comparison of the individual $\sigma_{pt}$ values shows the good performance of the 1:1/1:2 model.** In the **2:1/1:1/1:2 model**, even the correct lg $K_{12}$, that is connected to acceptable results in the Monte Carlo simulation, has higher $\sigma_{pt}$ values compared to those obtained for the correct model. The analysis applied to incorrect parameter lead to larger $\sigma_{pt}$ values in general.

the model from **1:1/1:2** to **2:1/1:1/1:2** (0.25491), while $\sigma_{pt}$ for lg $K_{21}$ increases from 0.25578 by a factor of 1.8 to 0.45884. Removing the cut-off, all partial standard deviations get worse and the differences between the models become less clear. Comparing the different sets of $\sigma_{pt}$ for both cut-off ratios (1.8 and 2.0) no qualitatively differences can be observed. Slightly lower $\sigma_{pt}$ values are obtained for the correct model with the higher cut-off ratio, which is comprehensible since having the lower cut-off ratio, data points are included, at which the saturation concentration for 1:2 complex has not been reached yet. The trend in the partial standard deviations is similar to the trend of Monte Carlo results, where the lowest values of H(x) and $\sigma_{dist}$ are obtained for the correct model. Differences occur in the **2:1/1:1 model**, where as result of the Monte Carlo simulation both the entropy and $\sigma_{dist}$ of the parameter lg $K_{11}$ are lower than of the parameter lg $K_{21}$ while using the reduction analysis, the resulting $\sigma_{pt}$ is higher for lg $K_{11}$. Remarkable is the "visual" detection of the insufficient **1:1 model** as the removal of data points lead to an increase of the best-fit parameter, each below and above the optimal ratio of 1.0. Even in the **2:1/1:1/1:2 model** lg $K_{11}$ exhibits a more constant behaviour.

## Data set with 1:1 stoichiometry

In analogy to the data set based on 1:1/1:2 model, a simulated NMR titration with an underlying 1:1 model was generated and tested using Monte Carlo simulations, Cross Validation and Reduction Analysis. The main results, covering the Shannon entropy of the distribution of individual parameters as well as the partial standard deviation are shown in Fig. 9. A summary of all results is presented in the supplementary material (Figs. S1–S5). The matrix decomposition analysis reveals two factors, which is in agreement with two species contributing to the observed signals (Fig. S20A). Extreme outliers were obtained during both Monte Carlo simulations on top of the best-fit **2:1/1:1 model**. Since the Shannon entropy is calculated using 30 bins to partition the histogram and due to the presence of the outliers which enlarge the bins, a reasonable H(x) value for lg $K_{11}$ was not obtained. The standard deviation is independent of the partition of the histogram, therefore reasonable results can be obtained regardless the distribution. While the combination of

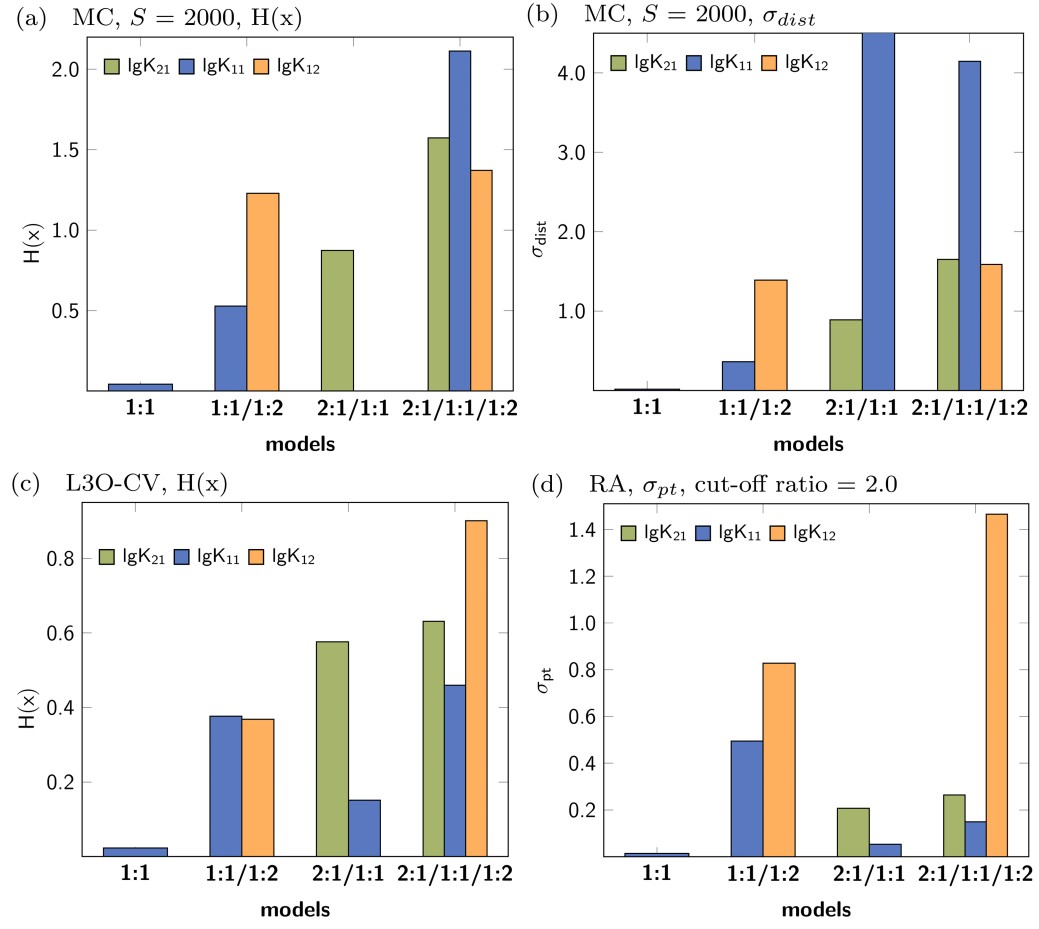

**Figure 9** **Calculated statistical descriptors for four models (A–D) obtained after Monte Carlo simulation, Leave-3-Out Cross Validation and Reduction Analysis performed on a simulated NMR titration with a 1:1 model.** Due to the presence of extreme outliers after Monte Carlo simulation on top of the best-fit **2:1/1:1 model**, the Shannon entropy could not be calculated for lg $K_{11}$ in (A).

the **2:1/1:1 model** and Monte Carlo simulations lead to distribution with high standard deviation, the obtained individual parameters after Cross Validation don't exhibit that pattern. According to the obtained statistical descriptors for each parameter, the **1:1 model** describes the simulated titration data best. However, the parameter lg $K_{11}$, which performs very well in the **1:1 model**, performs badly if more **complex models** are analysed using Monte Carlo simulation or Cross Validation.

## Data set with 2:1/1:1 stoichiometry

The collection of the statistical descriptors obtained for analysis of the simulated data set with 2:1/1:1 model are compiled in Fig. 10. A summary of all results is presented in the supplementary material (Figs. S11–S15). The factor analysis indicating two species (Fig. S20C) is not in agreement with the original model using three species contributing to the observed shifts. In contrast to the 1:1 model, where only one parameter was correct, in the current example both, lg $K_{21}$ and lg $K_{11}$, are mandatory to describe the data.

According to the results visualised in Fig. 10, the descriptors indicating the performance of the parameter of the **2:1/1:1 model** are better (lower) than in the **2:1/1:1/1:2 model**. Furthermore the descriptors indicate that lg $K_{11}$ performs in the **2:1/1:1 model** better than in the **1:1/1:2 model**. However, in contrast to the previously discussed data of a simulated 1:1/1:2 experiment, the values of the descriptors are in general higher, although the absolute values are of limited meaning. The differences of the descriptors for lg $K_{11}$ in either the applied **1:1** or the **2:1/1:1 model** are not as clear as between the applied **1:1** and the applied **1:1/1:2 model**. In case of H(x) for lg $K_{11}$ after MC with 2000 steps, the entropy drops from 0.08374 (**1:1 model**) to 0.05235 (**1:1/1:2 model**) by a factor 0.63 in the simulated 1:1/1:2 titration and from 0.25172 (**1:1 model**) to 0.22620 (**2:1/1:1 model**) by a factor of 0.90 in the simulated 2:1/1:1 titration data set. Using the L3O-CV results, the difference is clearer. The entropy drops from 0.50499 (**1:1 model**) to 0.17749 (**2:1/1:1 model**) by factor 0.35 in case of the 2:1/1:1 titration. The change of the entropy in the 1:1/1:2 titration is by a factor of 0.68, from 0.04696 (**1:1 model**) to 0.03214 (**1:1/1:2 model**) respectively. However, the results in Fig. 10 indicate that the **1:1/1:2 model** performs worse than the **2:1/1:1** or **1:1 model**, therefore only **1:1 model** could be considered as alternative model. Comparing this results with results obtained for the pure 1:1 model, lg $K_{11}$ performs clearly worse if a **2:1/1:1 model** is fitted to a 1:1 data set. Following this pattern, the statistical post-process did not fail to detect 2:1/1:1 stoichiometry correctly, although the results are not as clear as in case of the 1:1/1:2 model.

### Data set with 2:1/1:1/1:2 stoichiometry

In contrast to the correct detection of the appropriate models in the previous discussion, the individual parameters obtained for the stability constants don't reflect lg $K_{21}$ as suitable parameter (Fig. 11). A summary of all results is presented in the supplementary material (Figs. S16–S20). According to the factor analysis, three species contribute to the observed signals (Fig. S20D), which is not in agreement with the data set. In both models, **2:1/1:1** and **2:1/1:1/1:2**, the statistical descriptors for lg $K_{21}$ show higher values than the descriptors of lg $K_{11}$ and lg $K_{12}$. Although the identification of the correct stoichiometric model was not possible with the statistical post-processing, this can be rooted back to the simulated data. The titration data was simulated with $lgK_{21} = 1.85$, $lgK_{11} = 3.60$ and $lgK_{12} = 2.40$, where only the stability constants lg $K_{11}$ and lg $K_{12}$ were recovered correctly (3.61 and 2.42). The stability constants lg $K_{21}$ was estimated to be 0.21, indicating no 2:1 stoichiometry at all. The correct estimation of 2:1 stoichiometry is somewhat difficult, as the concentration of the 2:1 species is lower than for the other species. Therefore the influence of this complex to the overall observed signal is of less importance.

### Averaged statistical descriptors

Averaged statistical descriptors for the stability constants calculated for each model and for each parameter are illustrated in Figs. 12 and 13. The chart in Fig. 12 shows the average of the descriptors like H(x) and $\sigma_{dist}$ obtained for all parameters within a single **model**. For example the blue bar represents the average of the descriptors for all parameters of the **1:1 model**, which is lg $K_{11}$. The orange bars represent the average of the descriptors

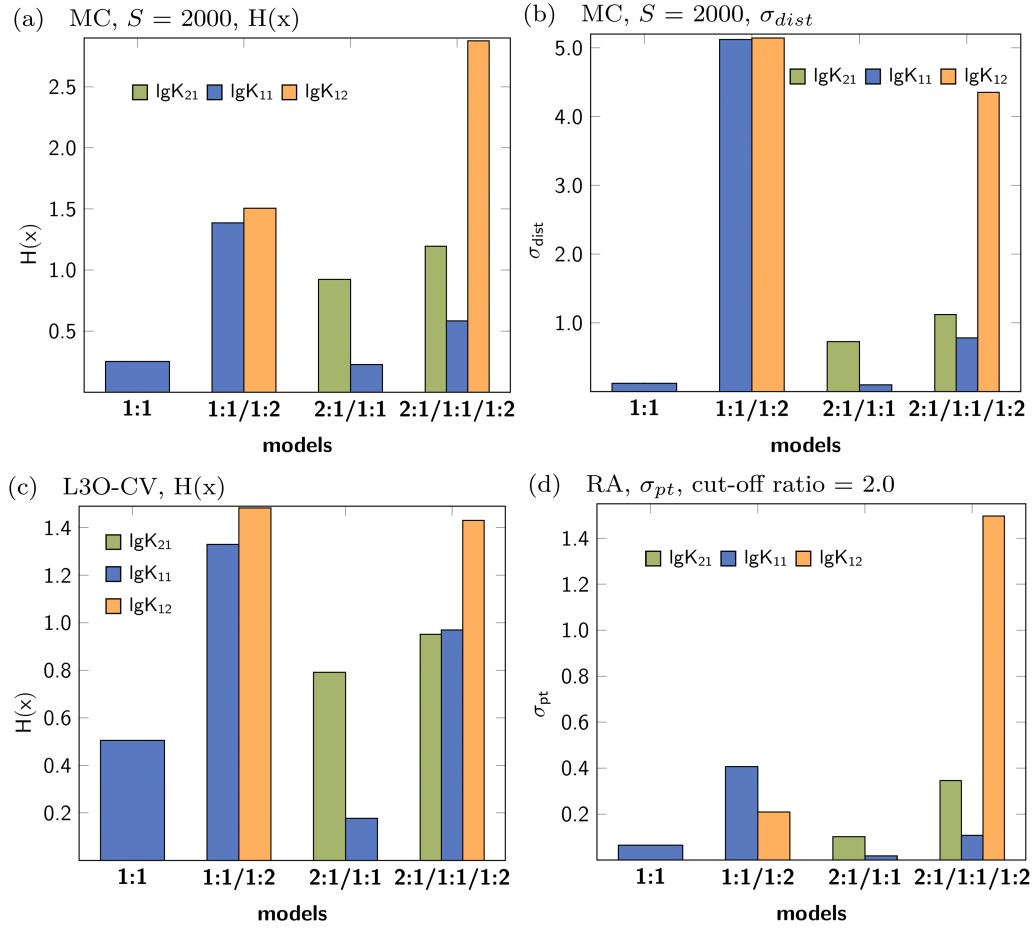

**Figure 10** Calculated statistical descriptors for four models (A–D) obtained after Monte Carlo simulation, Leave-3-Out Cross Validation and Reduction Analysis performed on a simulated NMR titration with a **2:1/1:1 model**.

for lg $K_{11}$ and lg $K_{12}$, which are the stability constants in case of the **1:1/1:2 model**. As each bar represents the average values of the descriptors for each **applied model**, the general quality of the fit using a special **model** on top of the data set is easily deduced. The model fitted best to the simulated 1:1 systems is the original **1:1 model**, where however the average descriptor is obtained for only one stability constant. The average partial standard deviation in case of the other **tested models** is higher, with the highest value obtained for the **1:1/1:2 model**. According to the average $\sigma_{pt}$, the 1:1/1:2 system is best described by the **1:1/1:2 model** and worst by the **2:1/1:1 model**. A similar agreement between the true simulated model and lowest averaged descriptors was obtained in case of the **2:1/1:1 model**, however the difference of the $\sigma_{pt}$ value compared to the **1:1 model** is very small. Using the average descriptors for $\sigma_{pt}$, the 2:1/1:1/1:2 model was not identified as the correct one. Better results were obtained fitting a **1:1** and **1:1/1:2 model** to the simulated data with 2:1/1:1/1:2 stoichiometry, which is in accordance with results discussed in Section 'Data set with 2:1/1:1/1:2 stoichiometry'. Results obtained with

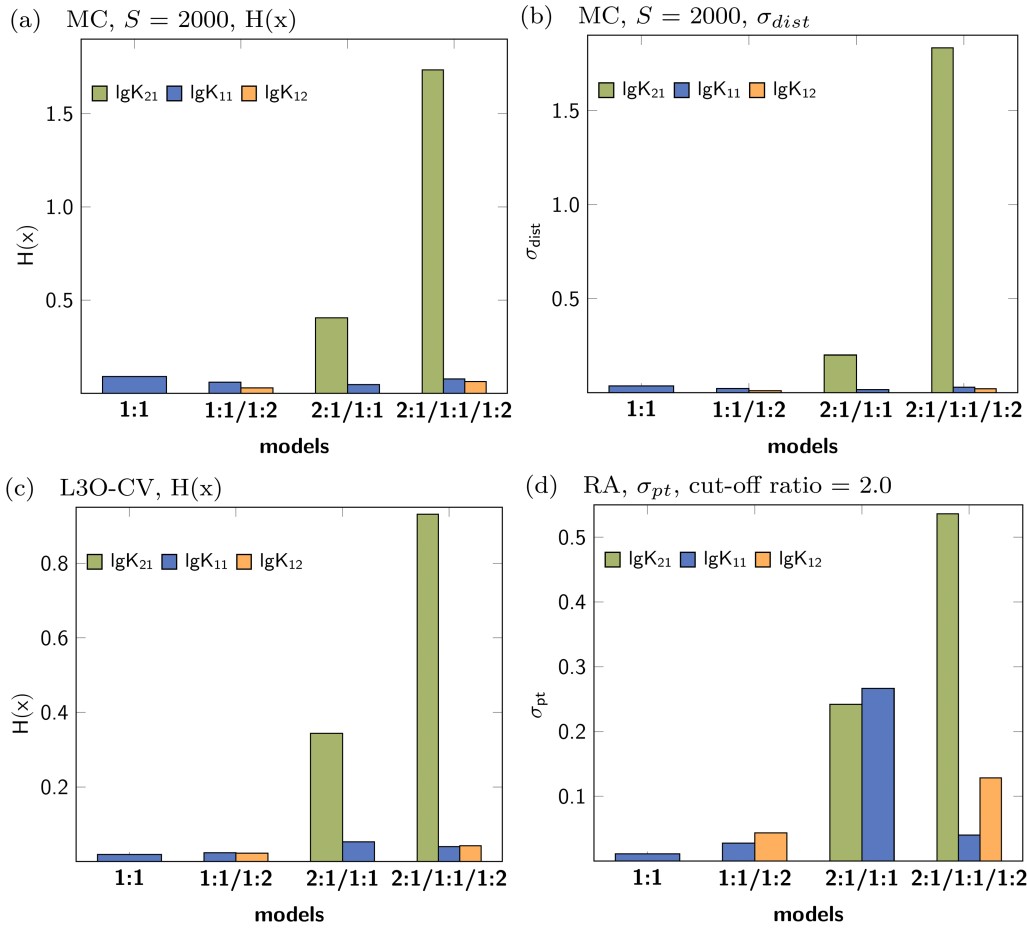

**Figure 11** Calculated statistical descriptors for four models (A–D) obtained after Monte Carlo simulation, Leave-3-Out Cross Validation and Reduction Analysis performed on a simulated NMR titration with a **2:1/1:1/1:2 model**.

Reduction Analysis were comparably recovered using Cross Validation and calculated Shannon entropy and standard deviation of the distribution. Comparing the averaged $\sigma_{pt}$ values with the averaged Shannon entropy, the difference is similarly small in case of the **1:1** and **2:1/1:1 model** applied to the 2:1/1:1 data set (Figs. 12A and 12B). In contrast, using $\sigma_{dist}$ to characterise the distributions of stability constants the standard deviation obtained for both, the **1:1** and **2:1/1:1 model** differ more clearly (Fig. 12C). The results with the remaining CV and MC calculations can be found in the supporting information. Apart from the mixed **2:1/1:1/1:2 model**, post-processing after RA, CV and MC identifies the correct models by assigning them the lowest value of the averaged descriptors compared to the remaining models. The absolute values of these descriptors have not been identified as meaningful on its own.

Similar to the average descriptors obtained for each model individually, the average can be calculated for the all parameters, *e.g.*, stability constants. In Fig. 13A, the blue bar represents the average of four $\sigma_{pt}$ values obtained for the stability constant lg $K_{11}$ from

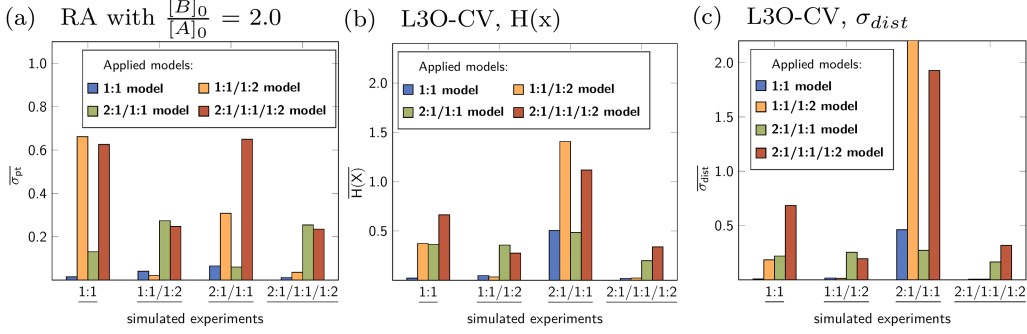

**Figure 12** (A–C) Model-wise average of the statistical descriptors calculated for each model using the model parameters for a given simulated titration experiment. In case of the fitted 1:1/1:2 model on top of the simulated 2:1/1:1 experiment, a value $\sigma_{dist}$ around 11 was obtained.

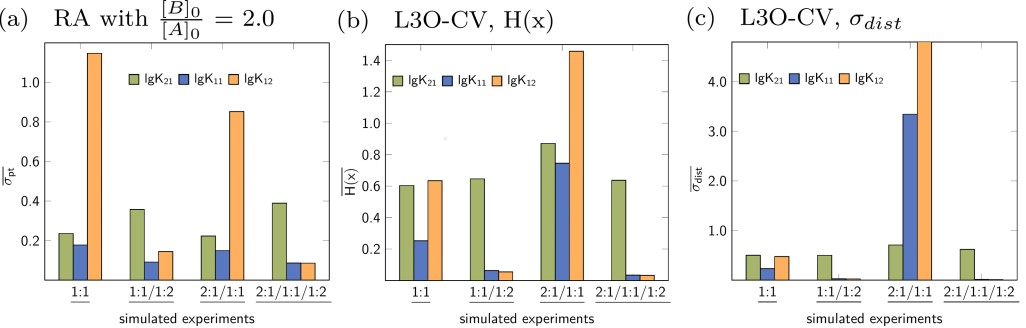

**Figure 13** (A–C) Parameter-wise average of the statistical descriptors calculated for each parameter of all models applied to a given simulated titration experiment.

the **1:1**, the **2:1/1:1**, the **1:1/1:2** and the **2:1/1:1/1:2 model**. The green bar represents the average of all lg $K_{21}$ stability constants, however obtained for fewer models. The remaining average descriptors are obtained in a similar fashion for all stability constants of all data sets. It is obvious from the charts in Fig. 13 that apart from the **2:1/1:1/1:2 model**, the most appropriate stability constants in general have lower average values compared to those which are inappropriate. This is true for RA as well as L3O-CV using both descriptors. While this recovers the expectation derived by the previous discussion, there is a fundamental shortcoming in this context. In contrast to the average descriptors for each model, focusing on the stability constants or the other model parameters, an a priori knowledge of the number of model parameters is mandatory. Given the result in Fig. 13A for the 1:1 model, the difference between $\sigma_{pt}$ for lg $K_{11}$ and lg $K_{21}$ is much lower than compared to $\sigma_{pt}$ for lg $K_{12}$. On the other hand, in case of the 1:1/1:2 model, the difference between lg $K_{12}$ and lg $K_{21}$ is much lower than in the simulated 1:1 model. And the difference between the $\sigma_{pt}$ values for the correct stability constants lg $K_{11}$ and lg $K_{12}$ (1:1/1:2 model) is comparable to the difference between the correct lg $K_{11}$ and the inappropriate lg $K_{21}$ (1:1 model). Some differences become clearer if the CV results were analysed. However, if both, the number of

appropriate parameters and the parameter itself, are unknown, the parameter-wise average of the descriptors is not meaningful.

Based on the results for both strategies to calculated average descriptors, the meaningful approach is to average over all parameters for a given single model and not to average over all identical parameters for all models. However, both strategies are applied automatically upon comparing the results of the statistical post-processing for applied models in *SupraFit*.

## SUMMARY AND CONCLUSION

The application of Monte Carlo simulations, Cross Validation and Reduction Analysis to identify appropriate model parameters (*e.g.*, stability constants) and models in the analysis of NMR titration experiments was presented. Four different NMR titration experiments on the base of the models implemented in *SupraFit* (of 1:1, 2:1/1:1, 1:1/1:2 and 2:1/1:1/1:2 stoichiometry) were simulated. Each of the simulated models was analysed using the **four available models** and the best-fit results were further processed using statistical approaches. In the study, only the stability constants were discussed as they are the global parameters. The parameters linked to chemical shifts were not included as they represent only local parameter in a global fit analysis.

The visual inspection of the statistical results reveals for the histograms after MC and CV differently shaped distributions of the model parameters. Parameters coinciding with the stability constants that were used in order to generate the simulated data are associated with a narrower gaussian-like distribution of the possible parameters. Stability constants that were not part of the original model are connected with both, a broader and a non gaussian-like distribution, having sometimes several maxima. The breadth of the distribution is in agreement with the change in the confidence interval, that can be calculated based on the percentile method. To the best of the authors knowledge, the combination of Cross Validation and the percentile method has not been proposed as protocol to calculate confidence intervals.

Reduction analysis results are represented as function of the model parameter from the current ratio of the initial concentration of the components B and A ($\frac{B_0}{A_0}$), or in general from the last included input data point. In case of the correct model, the appropriate model parameters don't vary from the best-fit value in the same magnitude as the inappropriate model parameters upon removing the last data point. However, once a threshold is reached the difference to the best-fit value increases as well. In case of the NMR titration, this threshold coincides with the saturation point.

The rational non-visual comparison of the analysis is further done using statistical descriptors. The histograms representing the distribution of the individual stability constants resulting from Monte Carlo simulations and Cross Validation are the base for calculation of the Shannon entropy H(x) and the standard deviation $\sigma_{dist}$. The results of the Reduction Analysis are simplified by the descriptor $\sigma_{pt}$ which represents the deviation of any model parameter from its best-fit value above a threshold. The statistical descriptors were compared for (a) each stability constants individually, (b) averaged for all parameter in a tested model (model-wise) and (c) averaged over all identical parameters

covering all models (parameter-wise). The individual comparison of the stability constants was comprehensively discussed in case of the 1:1/1:2 model, showing clearly that the lowest value of the descriptors was obtained for lg $K_{11}$ followed by lg $K_{12}$. It was further observed, that lower values of that descriptor were obtained in Monte Carlo simulation if the input standard deviation was decreased. An analogous decrease of the values for lg $K_{21}$ in conjunction of Monte Carlo simulation with various $\sigma_{MC}$ values was not observed. Furthermore, the individual values for $\sigma_{dist}$ and H(x) were higher than for lg $K_{11}$ and lg $K_{12}$.

Monte Carlo simulation with different standard deviation as well as Bootstrapping and Cross Validation with different strategies to leave data points out result in distribution of the stability constants that indicate lg $K_{11}$ and lg $K_{12}$ as appropriate parameters, and therefore the original stoichiometry was recovered correctly. In the same fashion, Reduction Analyses result in $\sigma_{pt}$ values that for both threshold, 1.8 and 2.0, indicate both lg $K_{11}$ and lg $K_{12}$ as appropriate model parameters. Thresholds according to lower ratio would not be sufficient as the ideal saturation point in case of the 1:2 stoichiometry is 2.0.

Both strategies, parameter-wise and model-wise average descriptors coincide with the above outlined results indicating the 1:1/1:2 model as the best. The discussed protocol, including the averaged parameters, identified the 1:1 and 2:1/1:1 stoichiometry in the simulated data as well, however the descriptors associated with lg $K_{21}$ or the 2:1/1:1 model indicate worse performance compared to lg $K_{12}$ or the 1:1/1:2 model. This protocol however failed in the test case for a simulated 2:1/1:1/1:2 NMR titration. In case of the parameter-wise averaged descriptors, failure is intrinsic as from the three available model parameters the three best have to be picked. On the other hand, using the model-wise average parameters, the 2:1/1:1/1:2 model is best described by a **1:1/1:2 model**. However, this case is no contradiction as the best-fit **2:1/1:1/1:2 model** results in a lg $K_{21}$ value below 1. On the basis of the simulated experiment, no meaningful stability constant for a 2:1 species could be assigned, which therefore was correctly supported by the statistical post-processing.

While the outlined protocols indicate the potential to detect correct models and model parameters, the first shortcoming was identified in the correct detection of the 2:1/1:1/1:2 model. Another, more general, shortcoming arises as only one data set per model was tested. In case of the first deficit, it is however not clear if a model with as many parameters as occurring in the 2:1/1:1/1:2 model can be justified based on the available data points or if more data points have to be acquired. If there is however a strong evidence for a mixed 2:1/1:1/1:2 model which is not supported from results of the discussed protocols, the experimental set up may be changed. *SupraFit* provides an user interface, which help to plan the experimental set up, furthermore Vander Griend (*Kazmierczak et al., 2019*) provided a detailed study on the limits of the sensitivity with respect to the determination of stability constants.

Future research will therefore focus on how the experimental design has to alter in general, *e.g.*, include data points in various concentration regime, to then correctly estimate the more complex model. The failure to correctly detect the 2:1/1:1/1:2 model may coincide with the second shortcoming, that only one simulated data set was tested

during the proposed protocol. In case of another simulated titration experiment with 2:1/1:1/1:2 stoichiometry and reasonable best-fit binding constants for all species, the detection may not necessarily fail. Hence, the simulation framework in *SupraFit* will be extended and as a result, a large set of simulations can be performed and the limits of the proposed protocol can be analysed in more detail. This will also include the sensitivity with respect to the number of NMR signals included and the absolute change of the NMR shifts observed. Furthermore, with the recently implemented models that may have any possible species $A_aB_b$, the protocol will then be tested with more species available. The extension of the simulation framework will also include adding noise to the independent data, hence the influence of concentration errors in stock solution can be accounted for. The application of the protocols or at least parts of them to ITC experiments is as well a topic of future research. It is, however, assumed that the current status of the protocol, that can be used with the stable version 2.0 of SupraFit, will already help to improve the analysis and interpretation of NMR titration and underpin the potential of the statistical approaches which account for models which are nonlinear in the parameters.

## ABBREVIATION

Abbreviations can be found in Supplemental Information 1.

## ACKNOWLEDGEMENTS

The author thanks Prof. M Mazik, TU Bergakademie Freiberg for her support as well as Dr. Sebastian Förster, Dr. Stefan Kaiser and Dr. Felix Amrhein for finding bugs and constructive feedback during the development of *SupraFit*. Dr. Anke Schwarzer and Dr. Manuel Stapf are thanked for proofreading and helpful comments on that manuscript. The reviewers are thanked for their constructive input on that topic and manuscript.

### Funding

This work was supported by the Centre of Advanced Study and Research - Freiberg (GraFA) and the Saxonian Ministry of Science, Culture and Tourism (SMWK)(project number 100333374). The funders had no role in study design, data collection and analysis, decision to publish, or preparation of the manuscript.

### Grant Disclosures

The following grant information was disclosed by the author:
The Centre of Advanced Study and Research - Freiberg (GraFA) and the Saxonian Ministry of Science, Culture and Tourism (SMWK)(project number 100333374).

### Competing Interests

The author declares that he has no competing interests.

## Author Contributions

- Conrad Hübler conceived and designed the experiments, performed the experiments, analyzed the data, performed the computation work, prepared figures and/or tables, authored or reviewed drafts of the article, and approved the final draft.

## Data Availability

The raw data is available at Zenodo: Hübler, Conrad. (2022). Raw data for 'Analysing binding stoichiometries in NMR titration experiments using Monte Carlo simulation and resampling techniques' [Data set]. Zenodo. https://doi.org/10.5281/zenodo.6539577.

The SupraFit code is available at GitHub:

https://github.com/conradhuebler/SupraFit.

## Supplemental Information

Supplemental information for this article can be found online at http://dx.doi.org/10.7717/peerj-achem.23#supplemental-information.

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
