# Peer review of "Analysing binding stoichiometries in NMR titration experiments using Monte Carlo simulation and resampling techniques"

_PeerJ Analytical Chemistry, doi:10.7717/peerj-achem.23_

## Round 0.1 · original submission · Major Revisions

Dear Prof. Hübler,

Please, find enclosed three strong reviews from experts in the field, which all agree that there is merit in the publication, but it needs a significant revision.

I am looking forward to receiving your revised manuscript.

·

Basic reporting

This manuscript addresses the problem of fitting NMR titration curves to complex binding equilibria with the coexistence of liganded species corresponding to different stoichiometries, such as 1:1 (AB), 2:1 (A2B) and 1:2 (AB2), to extract the thermodynamic constants and the stoichiometry information. Fitting of stoichiometric models involves non-linear equations containing multiple parameters, which makes it a difficult fitting problem. This approach is directed to the study of small molecule complexes, when the binding reaction is in fast exchange at the NMR time scale, using the program SupraFit that was recently published. Synthetic NMR titration curves were generated assuming each one out of four binding models: 1:1, 1:1/1:2, 1:1/2:1, 2:1/1:2/1:1. Each synthetic curves was separately fitted using SupraFit and assuming all four binding models to understand whether the model used to generate the data could be correctly identified. Three different approaches were tested to evaluate the fitting and the program's ability to choose the correct model and the correct binding constants: Monte Carlo simulations, cross validation, and reduction analysis. Their results indicated that when the binding constants are correctly fitted, the distributions are relatively narrow, and that helps to select the proper binding model. Reduction analysis also seemed to be sensitive to the correct model. Non-visual analysis was carried out using statistical descriptors of the histograms resulting from Monte Carlo and Cross Validation calculations, such as the histogram standard deviation and the Shanon entropy. These descriptors seem to be sensitive to the different models (1:1, 1:1/1:2, 1:1/2:1 models), but they were unable to identify the more complex 2:1/1:2/1:1 model that is anyway a more difficult task. Overall, the proposed metrics have the potential to pinpoint the correct model. However, the paper would benefit from being more concise. For example, part of the detailed discussion fittings of all four titration curves (four models) could be left out to give more emphasis to the 1:1 and the 1:1/1:2 models, and on the model-wise average statistical descriptors (Fig 12). In addition, more detail information about the binding models could be given in the methods section. I understand that the binding constants are being expressed in terms of log10, however, it would be interesting to see the units, at least for the chemical shifts shown in Figure 1.

Experimental design

This manuscript main question is clearly stated. The experimental approach is based on the analysis of synthetic NMR titration data using the program SupraFit. In this sense, the manuscript seems to be useful a guide for the potential users of that program and those who need to analyze complex binding equilibria.
The methods are described with sufficient detail provided that additional information on the binding models are given.

Validity of the findings

The main results are discussed in very detail. But it will become much easier for the readers if the text is shortened.

Reviewer 2 ·

Basic reporting

Hubler presents here analysis of binding stoichiometries comparing a number of resampling methods.
This is a timely work, addressing a difficult (impossible?) challenge in supramolecular chemistry, which is model selection. Hubler applies MC and other resampling methods to tackle this challenge. While successful with some of his simulated datasets he admits it does not quite work on his most complex ones 2:1/1:1/1:2. I would go further and point out that although the results are promising for the other models tested, they are only tested on one particular combination of association constants (e.g. the 1:1/1:2 seems to be a weakly negatively-cooperative systems). Further, the MC and other resampling methods only seem to consider NMR measurement errors (chemical shifts) but not concentration errors.
That all said, I think the paper has great value and should be published provided a number of issues are addressed :

1. The authors recognises the above mentioned limitations in the manuscript text itself.

2. It is a bit hard to follow the difference between “models” and “data”. Could it be made clearer in each case which “raw” data set is being discussed? Maybe label the raw data set in bold like 1:1, 1:1:/1:2 etc to make the differences between data and models clearer.

3. To help the reader, please add lg (presumably log10?) to the abbreviation list.

4. Line 53 on page 2 – The criticism of Job’s Method was first systematically outlined in J. Org. Chem., 2016, 81, 1746 and then expanded on by Hibbert and Thordarson 2016. Combined those two papers didn’t just criticise Job’s Method but unequivocally showed Job’s Method to be wrong. The text needs to make it a bit clearer that Job’s Method simply does not work.

5. This reviewer will admit not knowing much about Shannon entropy. But two question arises regarding the overview given on page 4: For systems that follow normal (gaussian) distribution, does the Shannon entropy really give the experimental any further insight into the system than you would get with conventional descriptors such as the mean and the standard deviation? And I got a similar question for systems that are not normally distributed? Could you not just also describe them in terms of the 2.5% and 97.5% percentile + the mean? Wouldn’t that give a similar insight into the broadness of the peak but perhaps make more sense to the experimentalist than the concept of Shannon entropy which is not well known in chemistry?

6. Line 193 page 7. Just double checking but is the 0.001 delta(MC) only on the ‘y-data’ – and does it then mean an absolute uncertainty of 0.001 ppm in the NMR data? Or is it 0.1% relative error?
And what about concentration errors/noise? The Hibbert-Thordarson 2016 showed that errors on host and guest concentrations have a significant effect. Under real-world conditions those errors (making up stock solutions, syringe reproducibility etc…) cannot be ignored.

Experimental design

See above

Validity of the findings

See above

Additional comments

See above

·

Basic reporting

In general this is well-written with good explanations of many abstract mathematical ideas.There are however several spots that should be rewritten to decrease ambiguity: "arising the limitation (118)", "as essential (181), "somehow comparable (181)".
Also there are several instances where the term data points are used, when in fact data columns might be correct (148 & 340).
Also there are several instances of commas placed in an inappropriate point in a sentence: 316, 355, 396.
And too many significant figures on table 1.

The author seems familiar with the work of Thordarson but not the work of Vander Griend. Three papers in particular that should be assessed, and the results of which integrated with this manuscript are:
(1) Kazmierczak, N. P.; Chew, J. A.; Vander Griend, D. A. A Reliable Algorithm for Calculating Stoichiometry Parameters in the Hard Modeling of Spectrophotometric Titration Data. Journal of Chemometrics, e3409. https://doi.org/10.1002/cem.
(2) Kazmierczak, N. P.; Chew, J. A.; Vander Griend, D. A. Bootstrap Methods for Quantifying the Uncertainty of Binding Constants in the Hard Modeling of Spectrophotometric Titration Data. Analytica Chimica Acta 2022, 339834. https://doi.org/10.1016/j.aca.2022.339834.
(3) Kazmierczak, N. P.; Chew, J. A.; Michmerhuizen, A. R.; Kim, S. E.; Drees, Z. D.; Rylaarsdam, A.; Thong, T.; Laar, L. V.; Griend, D. A. V. Sensitivity Limits for Determining 1:1 Binding Constants from Spectrophotometric Titrations via Global Analysis. Journal of Chemometrics 2019, 33 (5), e3119. https://doi.org/10.1002/cem.3119.

More details on how these papers should be considered in the next section.

Experimental design

While it is worthwhile to help people understand how to differentiate between various models when fitting spectroscopic data, one of the first analyses that should always be done is a simple singular value decomposition (SVD) of the data to try to ascertain the number of distinct chemical species in the model. This is the first defense against using a model with too many species. This manuscript should definitely include an SVD analysis of each dataset. I am interested to see how the value of the RA analysis changes in light of the SVD analysis.

Presumably, this manuscript is limited to the 4 models showcased because of the limitations of the program being used. Perhaps I am wrong. There are programs which are not limited to models with analytical solutions (e.g. Sivvu). Please provide a rational for these chosen models, especially given that they vary in size from one to three complexes, which makes them less than ideal for comparison.

Vander Griend et al. have previously shown the usefulness of bootstrapping on the data to ascertain the uncertainty of the binding constants. It is very similar to both MC and CV and is arguably superior to both, if only because no a priori assumptions on error are required and the resulting data replications are identical in size to the original. Furthermore, bootstrapping has been tested and calibrated against not only signal error, but also transmittance, concentration and stock solution error. it also been tested on experimental data.

Vander Griend et al. have also developed an algorithm for searching out models so that the model is no longer necessarily an input. Within their work are numerous insights into the relationship between model and data that should be considered with this manuscript.

Validity of the findings

Please provide more details in the manuscript for the construction of the artificial datasets: lgK, [A]0, [B]0, number of equivalents. These parameters are crucial for appreciating the dataset and the binding regime to which it belongs. Also, Vander Griend et al. have provided insights into which chemical solutions contain the most information about a binding constant and consequently how to redesign a titration to best differentiate alternate binding modes. This seems especially relevant to some of the discussion in the final paragraphs of the manuscript.

Additionally, it would useful if computational time statistics were provided. Computational time and resources is mentioned in line 149, but this should not be used as an excuse. 1000's of replications can be run on much bigger spectrophotometric datasets in a matter of minutes on a desktop machine now with small models like those in this manuscript.

Additional comments

This manuscript reminds me of my first manuscripts in this area, so i hope my review comes across as constructive. Overall i see potential here, particularly in the RA, which is the more novel component to my knowledge.

Clearly this reviewer is not hiding my identity. I am very familiar with this type of work, having written my own code and worked for years to publish in this area. it is a delicate balance to publish original results and promote one's own computer program. I welcome the author reaching out to me directly if they so choose (for advice and dialogue, but not badgering please).

BTW, It looks like all your code is ultimately accessible, which is commendable.

Also, FYI, Thordarson has a website for NMR fitting as well called supramolecular.org. And the website sivvu.org was designed for UVvis data but works for NMR data as well.

---

## Round 0.2 · Major Revisions

Dear Authors, Thank you for your reviewing efforts. One of the reviewers is still pointing out major issues, despite the other two reviewers being ready to accept for publication.
As the third reviewer is an expert in the field, I would like to see all of their points answered positively or (not and justify why).

·

Basic reporting

Overall, the author addressed my comments on the initial version of the manuscript. Although the text was not shortened, this revised version was clearly improved by stating which models were used to generate the synthetic data (underlined) and which models were used to fit the data (in bold). When appropriate, comments to compare with the performance of other programs to analyse NMR titration data, and additional explanation on the Shannon entropy, were added in response to the other reviewers, and that also helped to improve the text. Occasionally, the new (blue) text showed some typos that should be corrected, for example lines 151-152 on page 4. Furthermore, the variables used in Eqs 3 and 4 should be defined.

Experimental design

As written before, this work was well designed to evaluate the performance of different statistical descriptors to identify the correct stoichiometric model in NMR titration experiments using the program SupraFit. The validity and pitfalls of the proposed metrics were extensively discussed.

Validity of the findings

The validity and the pitfalls of the approach were well discussed. Overall, I agree with the author that one way to improve the fitting of more complex models such as the 1:1/1:2/2:1 model could be to simultaneously fit titrations made in different concentrations ranges.

Reviewer 2 ·

Basic reporting

The author has in my opinion addressed all the key questions that this reviewer has. Moreover, with some of the additions to the paper in response to the other reviewers, the paper has been strengthened further. I note that the reviewers comments and the authors reply raise a lot of interesting questions for the future, e.g. the subtle but important differences between UV-Vis and NMR data, is boot-strapping the best method (?), symmetric / non-symmetric error distribution and how to report it (Shannon? 2.5% - 97.5% gaps etc) and can uncertainty on concentration be ignored? This though will not be solved in this paper but the paper is nevertheless an important stepping stone towards settling some of these questions and should therefore be published now without further delay.

Experimental design

no comment to add.

Validity of the findings

no comment to add

Additional comments

no comment to add

·

Basic reporting

I find figure 1 and its caption to be quite confusing. what is "histogram like"? Also, 'after' is an odd choice of preposition here. finally, the first sentence of the caption essentially point to a 'simulation of a best-fit model'. Fit to which data?

Experimental design

SVD analysis, including evolving factor analysis, should be foundational to the quest of the paper to evaluate how various models, with differing numbers of species, potentially fit various datasets.

Validity of the findings

Thank you for employing SVD as a tool. I think it is the key tool for determining the number of factors. there is no need to cite SIVVU as SVD is a simple mathematical analysis of data. I disagree that it is not meaningful for small datasets like NMR. if there are signals present that can support a model, then SVD is meaningful by definition. Furthermore, if SVD reveals that there are three factors in the data, then testing models with a different number of factors is unproductive. Finally, it is incorrect to claim that "no further information about stoichiometry of the species can be deduced using the analysis (line 329)." Evolving factor analysis can be used to further pinpoint the location of the factors within the dataset which is turn points to their stoichiometry values.

Unfortunately, i think that the findings of this manuscript need to be thoroughly re-evaluated and re-organized given the helpful insight of SVD. It is the first tool to ascertaining the correct model in virtually every such situation. Applying models with the wrong number of chemical species can be ruled out at the start. Simply mentioning it and partially dismissing its applicability is inadequate.

Additional comments

In the debate of bootstrapping vs. monte carlo vs. CV vs. LXO, it would be valuable to present the results of each side by side. bootstrapping, particularly on the columns of the dataset as well as the residuals, has been calibrated using monte carlo simulations (Kazmierczak et al 2022). Until the others have, there is not much point in arguing.

---

## Round 0.3 · accepted · Accept

Dear Author,

Thank you for addressing the reviewers' comments in two review rounds.

I have assessed the revision myself, and I am happy with the current version. Therefore, I see that your manuscript is ready for publication.